# Vitellogenins Level as a Biomarker of the Honeybee Colony Strength in Urban and Rural Conditions

**DOI:** 10.3390/insects16010025

**Published:** 2024-12-29

**Authors:** Łukasz Nicewicz, Agata Wanda Nicewicz, Mirosław Nakonieczny

**Affiliations:** University of Silesia in Katowice, Faculty of Natural Sciences, Institute of Biology, Biotechnology and Environmen-tal Protection, Research Team of Animal Physiology and Ecotoxicology, Bankowa 9, 40-007 Katowice, Poland; lukasz.nicewicz@us.edu.pl (Ł.N.); miroslaw.nakonieczny@us.edu.pl (M.N.)

**Keywords:** vitellogenins, honeybee, urban apiary, seasonal dynamics

## Abstract

Urban beekeeping is growing in popularity, but it raises questions about the quality and safety of bee products and the health of bee colonies in city environments. While the safety of honey and other products has been well-studied, bees’ health has received less attention. How does city life impact bees and their colonies? This study analyzed the strength of honeybee colonies and levels of vitellogenin, a protein vital for bee health and longevity, in worker bees from an urban apiary from May to August. Results were compared to those of a rural (agricultural) apiary. We found that urban bee colonies were more populous than rural areas. In addition, insects in urban colonies showed a unique mechanism: During periods of bee shortage in the colonies, the level of vitellogenins—proteins responsible for extending the bees’ life—increased in the tissues of workers. This compensation mechanism may help explain the evolutionary success of social insects like honeybees in diverse environments.

## 1. Introduction

Urban apiaries are currently quite common but, above all, trendy. It is practiced worldwide, on all continents except Antarctica [1]. Such an intensive development of urban beekeeping raises many questions about the justification and benefits of honeybee breeding in cities.

The urban ecosystem can be a suitable place for honeybees. It is characterized by a diverse food base available throughout the growing season thanks to the successive plantings of squares or flower beds. This is an undoubted advantage compared to agricultural areas, where crops bloom relatively short, and the crop-flowering intervals can be large [2,3]. In addition, in the city, bees are less exposed to pesticides than in agricultural areas [4]. It should be noted that this factor depends mainly on the policy of local authorities and the standards of urban greenery management.

On the other hand, the urban environment is characterized by constant, higher exposure to potential stressors compared to agricultural areas (high temperatures, anthropogenic pollution, complex spatial structure) [5,6,7,8,9]. These factors can affect colonies for a long time at a level that causes a no-observed-adverse-effect level (NOAEL). However, their simultaneous occurrence can cause multistress [10]. Moreover, while bee mortality is relatively easy to estimate, sublethal exposure to the above-mentioned factors can subtly change bee physiology. Hence, a multifaceted approach is necessary to assess the impact of bees functioning in urban ecosystems. It seems that the complexity of the issue of bee welfare in urban apiaries is a key reason for the small number of studies on this topic.

To our knowledge, only two studies have been published so far that directly [8] or indirectly (through the productivity of bee colonies [7]) assessed the condition of bee colonies. Lecocq and colleagues [7] revealed that bee colonies from urban areas are more productive than colonies from suburban and rural areas. Nicewicz et al. [8] showed that honeybees from urban apiaries are under stress factors, other than those from rural areas, inducing stress response in foragers, e.g., the Hsp70 synthesis induction. Thus, the living environment of bees is not neutral to the condition of these insects at the individual level and probably for the entire bee colony. Hence, as the authors highlighted, there is a need for further research. Therefore, our study aimed to check whether different exposure to stress factors in urban and rural apiaries affects the strength of honeybee colonies. We also monitored the vitellogenins level in various tissues of honeybee foragers from these apiaries.

Vitellogenins (Vgs) are phospholipoglycoproteins present in almost all species of oviparous animals, acting as reserve material for the developing embryo [11]. In honeybees, however, the role of Vgs has expanded to include non-reproductive aspects. Vgs have been demonstrated in all castes, but their tissues concentration depends on the current function in the bee colony [12,13]. The role of Vgs in the bee body is pleiotropic. They play a crucial role in protection against oxidative stress and enhance longevity in queens and workers, not only long-lived winter bees [14,15,16]. Moreover, Vg is also linked to honeybee immunity in queens and in worker bees. The *Vg* gene expression level increases after wounding injury, and the protein binds to necrotic cells. The Vg protein recognizes bacteria and fungi and facilitates their phagocytosis. This protein is a zinc ion donor for hemocytes to provide their immune function. Moreover, the Vg is a carrier of immune memory from one generation to the next in the honeybee [17,18,19]. Therefore, the Vg level can be assimilated as an indicator of the overall health and long lifespan [20]. Vitellogenin is helpful as a biomarker of exposure to estrogenic compounds in aquatic invertebrates [21]. Recently, the possible use of Vgs as a biomarker of honeybee exposure to insecticides has been indicated [22]. Therefore, we would like to check the possibility of using the Vgs level as a biomarker of bee welfare in urban/rural apiaries. Thus, here, we analyzed the vitellogenins in various tissues in foragers during the season May–August in the urban apiary compared to the apiary from the agricultural area (reference apiary).

## 2. Materials and Methods

### 2.1. The Experimental Setup

All experiments were conducted on European-derived Carniolan honeybees (*Apis mellifera carnica* Pollman, 1879). The insects were collected from two apiaries that were specially established for the experiment. Experimental setups were based on the global guidelines for toxicological research in bee colonies [23,24,25]. The apiaries were established in August 2017, and 6-frame nucs with inseminated queens were obtained from a certified Polish breeder. Bees were kept in 10-frame wooden hives with a hygienic bottom. Tests and material collection were carried out 9 months later to provide the acclimation period.

Both apiaries consisted of six colonies. The first apiary was a typical urban apiary located on the roof of the Faculty of Law and Administration of the University of Silesia in Katowice (Katowice Upland, Silesian Voivodeship, 50°15′37.658″ N 19°1′56.707″ E). The apiary was located at a height of 30 m above ground level in the centre of Katowice, next to one of the busiest roads in Poland—National Road No. 86 (fourth place among the busiest roads in the country with an average daily annual traffic of 112,736 vehicles/day, [26]). The rural apiary (as a reference location) was located in a typical agricultural area on the Wieluń Upland, in the Lipie commune, in the village of Parzymiechy (Silesian Voivodeship, 51°2′29.462″ N 18°45′28.981″ E). The area is mainly devoted to extensively cultivating rye, buckwheat, and potatoes. The apiary was located at ground level next to the forest. The nearest city (Krzepice) is 15 km away, and the closest potential pollutant emitter—the Liberty Częstochowa steelworks and rolling mill—is 50 km away. The distance between the apiaries was 120 km.

Each apiary consisted of six families marked with consecutive numbers (one to six) and a symbol of the apiary location (U—urban, R—rural). The number of bee colonies in each apiary was determined based on recommendations for bee toxicological tests [25]. Colonies in all apiaries contained young queen bees of the same age. Each colony from the urban apiary was permanently linked to one of the colonies from the reference apiary, forming pairs marked with the same number. Each pair contained queen bees—sisters from pure breeding lines, artificially inseminated with semen from the same pool of drones. Consequently, six pairs of sister colonies were divided into two types of apiaries (urban or rural; Appendix A). This allowed for minimizing the variability of the obtained results, resulting from the genetic variability of the bees.

### 2.2. Bee Colony Strength Estimating

The strength of colonies was assessed by estimating the population of adult honeybees using the subjective method of Delaplane and colleagues [27], modified for a 10-frame hive. This method is recommended for analyzing the bee colony in ongoing field experiments.

This method consisted of a subjective assessment of the degree of bee coverage of each frame in the hive. Visual estimates of bees on combs were always carried out by two researchers simultaneously. Each colony was opened, and combs of bees were sequentially removed. Each researcher visually estimated the percentage of the comb surface covered by bees and expressed it as a percentage, where 0% means no bees on the frame, and 100% means so-called sitting on black. Each comb was assessed from two sides (A and B). Then, the results for these two researchers were averaged for each side of the combs. The same two researchers always assessed colonies to exclude the factor of the observer’s variable perception of the strength of the colony. The number of bees on each side of the comb was calculated based on the data about the surface area of each side of the frame and expected bee density when worker bees fully occupied the frame. The result was then summed to obtain the total number of bees present in the hive on the combs.

The bee colony strength was assessed monthly from May to August 2018 in both apiaries (Appendix A). The assessment was made at the beginning of each month (from May to August) in similar weather conditions (sunny days, slightly cloudy, air temperature of at least 22 °C, little wind) and at the same time of day (between 3:00 p.m. and 5:00 p.m.) in both apiaries [27].

### 2.3. Vitellogenins Level Determination

Foraging honeybees after 21 days of life were used for the Vgs concentration determination (Appendix A). Insects were collected using tweezers directly from the hive entrance. Only bees returning to the hive with pollen loads or nectar/water in the crop (distinctive through distended abdomens) were collected to avoid the guard bees.

The concentration of vitellogenins (Vgs) was analyzed in the brain, fat body from all body tags (necessary due to the functional division of this tissue [28]), and the entire bodies of foragers. After being prepared on ice, the tissues were weighed (the scale reading uncertainty ± 0.0001 g) and homogenized in 1 M phosphate-buffered saline (PBS; pH 7.4) (BioShop) (100 μL PBS/10 mg tissue). All homogenates were centrifuged (3000 rpm, 20 min, 4 °C). The supernatants were decanted, and the aliquots were kept in Eppendorf microtubes at −70 °C until the measurements were performed. The number of individuals from which tissues/organs came and constituting one research sample was selected experimentally. One whole bee, one brain, and fat bodies from three bees per sample were used for these analyses. Six samples for each tissue were prepared. To sum up, 24 bees from each colony and each apiary were collected in each term. In each term, 144 bees were collected from each apiary. In total, 576 individuals from each apiary were used for analysis.

A commercially available diagnostic kit (Honeybee Vitellogenin (VG) ELISA Kit, MyBioSource, San Diego, CA, USA) was used for the determinations. Vgs levels were determined in the tested tissues using the sandwich ELISA kit with the HRP colorimetric detection system to detect Vgs antigen targets in samples by applying all procedures of the commercial kit procedure. The concentration of Vgs in homogenates was calculated based on a standard curve determined based on standard solutions included in the kit and expressed in ng Vgs mg tissue^−1^. The negative control was the reaction mixture in which the homogenate was replaced with sample dilution buffer. Six replicates were prepared as research samples from each bee colony from both apiaries. Each sample from each tissue was analyzed in three repetitions, from which the average for the research sample was taken (total observation for each tissue = 18). The reference group included tissues/organs of foragers from a rural apiary.

### 2.4. Statistical Analysis

The data were expressed as means ± SD. The normality of distribution was checked using the Kolmogorov–Smirnov and Shapiro–Wilk tests with a confidence interval of *p* ≤ 0.05 and homogeneity of variances using Leven’s test. Whenever a significant effect between foragers from urban and rural apiaries was observed, the Tukey multiple comparison test was used for a post hoc one-way analysis of variance (ANOVA). Sidak multiple comparison tests with correction for multiple comparisons were used to analyze differences between colonies from the same apiary and the same colony in various months. The Pearson correlation was used to determine the relationship between colony strength and the Vgs level in various tissues. Results with *p* ≤ 0.05 were considered to be significant.

Analysis of variance for the colony size and the Vgs level from both apiaries was performed using MANOVA analysis using the type of apiary (urban vs. rural), the type of analyzed families (marked with colors), the date of sample collection, and the analyzed tissue as sources of variability were made using multivariate analysis of variance (MANOVA test, *p* ≤ 0.05).

The data were analyzed using GraphPad Prism^®^ ver. 8.1 (GraphPad Software Inc., San Diego, CA, USA) and presented in graphical and tabular form using the GraphPad Prism^®^ and Microsoft Excel ver. 2016.

## 3. Results

### 3.1. Colony Strength

The honeybee colonies strength, manifesting as the number of adult bees in the hive, were changed during the analyzed season—from May to August in both tested apiaries (Figure 1). In July, sister colonies U1 and R1 and U2 and R2 were the greatest-strength colonies. However, urban bee colonies were generally more numerous than their sister colonies from the reference apiary. For instance, in July, the U1 colony size from the city was twice as large as the sister colony from the reference apiary (R1). It was also shown that colonies from the same apiary differed in colony size during the analyzed season (Appendix A). The multivariate MANOVA analysis showed that the apiary location (*p* < 0.001), the date of bee collection (*p* < 0.001), and individual bee colonies marked with numbers (*p* < 0.001) were factors significantly differentiating the discussed results (Table 1).

### 3.2. Vitellogenins Level

Vitellogenins (Vgs) were revealed in all tested forager tissues/organs from urban and rural apiaries (Figure 2). The Vgs level fluctuated during the analyzed season—from May to August in all tissue/organs in both tested apiaries. The highest Vgs concentration was revealed in the entire body of foragers and the lowest in the brains of bees from both apiaries (even over six times higher in the R6 colony in July). It was shown that the statistically significant differences in the Vgs level in the entire body (Figure 2A), brain (Figure 2B), and the fat body of foragers (Figure 2C) between paired bee colonies from the urban and rural apiaries. There were also statistically significant differences in the Vgs level in all tested tissue foragers from the same apiary (Table 2 and Appendix A). Foragers from the reference apiary characterized higher levels of fat body Vgs. On the other hand, the concentration of Vgs was higher in the brains of workers from the urban apiary. Such clear relationships were not demonstrated for Vgs levels in whole-body foragers.

The MANOVA analysis indicated the significant influence of apiary localization, each bee colony, date of material collection, and type of tissue on the Vgs concentration in tissues/organs of foragers (df = 30; F = 48.9; *p* < 0,0000001) (Appendix A).

Statistically significant positive and negative correlations between the Vgs level in tested tissues within one colony and month of analysis were shown in Table 3. In addition, there were correlations between the Vgs level in tested tissues of foragers and the strength of colonies that were a source of these bees in both apiaries (Table 4).

## 4. Discussion

The strength of bee colonies is a parameter that indicates the impact of the environment—its weather, resources and pollution, the dynamics of pathogen populations, and beekeeping practices [23,29,30,31]. Here, we revealed that bee colonies from the urban apiary were more abundant than colonies from the rural apiary. It seems that the shaping of the bee colony strength depends primarily on environmental and not genetic factors because there were numerous statistically significant differences between sister colonies (urban vs. rural). Samuelson and colleagues [32] obtained similar results. Such factors that can impact the bee colony’s strength include abiotic factors (air temperature, exposure to pesticides, heavy metals, or other potentially toxic chemical compounds or mixtures) and biotic factors (food resources, their quality, parasites, pathogens, microbiota) [33].

Urban apiaries are often on the building’s rooftop, covered with materials that heat up quickly. For instance, the tested urban apiary was on a gravel roof. The ground under the hives gives off heat very slowly, and as a result, daily temperature fluctuations are less significant than in the case of a rural apiary. Additionally, appropriately positioning the hives on the roof towards the southeast allows colonies to be exposed to sunlight from the early morning hours. Such a location allows for earlier departures of foragers to the feeding bases because the activity of foragers is positively correlated with air temperature [34]. As a result, the foragers activity is extended during the day. A more widespread collection of pollen is then possible that can indirectly impact and regulate the number of broods raised and, consequently, influence the strength of the bee colony [35]. On the other hand, high rooftop temperatures will not support brood rearing in colonies because it requires a temperature ranging from 32 °C to 36 °C [36]. Therefore, the conditions in urban apiaries will be more complex because the bees will have to primarily cool the hive inside, resulting in higher energy expenditure [34].

In the study, the presence of Vgs was confirmed in the entire bodies of foragers, the fat body, and the brain (Figure 2). Vgs are synthesized primarily in the fat body of the bee abdomen and distributed to other tissues via hemolymph. Also, here, the presence of Vgs in the foragers’ fat body was demonstrated, regardless of the apiary location. The Vgs presence in the worker brains from both apiaries is the result of Vgs transport from the fat body of the abdomen to the brain glial cells, but also of autosynthesis in the head, where the cells of the fat body (impossible to separate) closely surrounding the brain can synthesize Vgs [22,37,38,39]. Therefore, the Vgs level in the fat body cannot be easily related to their concentration in the nervous tissue. Thus, the lack of simple correlations (only positive or negative) in Vgs levels between the brain and fat body is understandable (Table 3). Our results are confirmed by Münch’s team [38], which showed that the presence of Vgs in nervous tissue is isolated—increased levels of Vgs in the nervous system do not correlate with the levels of these proteins in other parts of the body.

It seems that the level of Vgs synthesis in the fat body and their concentration in the brain and entire body of foragers is affected primarily by environmental and not genetic factors. The presence of statistically significant differences in the Vgs level in all analyzed tissue between paired colonies (urban vs. rural, Figure 2) and colonies from one apiary (Appendix A) confirms the conclusion. Therefore, the variable level of Vgs during the May–August season was related to the current demand of the forager body for these phospholipoglycoproteins. Factors inducing oxidative stress can modify the Vgs concentration in worker bodies. For instance, the presence of Vgs in the forager brain may be a manifestation of such protective antioxidant mechanisms [38,40,41].

The Vgs synthesis level in the fat body, and thus indirectly their concentration in other forager tissues, can affect their nutritional status resulting from rich food bases [42,43]. Here, seasonal variability of Vgs concentration in the fat body of foragers may support this theory. The higher Vgs level in the fat body and the entire bodies of foragers from urban apiary than in rural apiary in 3 months of studies can indicate access to more abundant and diverse food bases in the city than in agricultural areas. Studies by Alaux and colleagues [44] confirm this conclusion. They showed that higher levels of Vgs were found in bees bred in semi-natural habitats with artificial plantings, abundant in cities, and not in agricultural areas with melliferous plants. Also, we revealed higher levels of Vgs in the foragers’ brains from the urban apiary than in bees from the rural apiary (Figure 2B). This may be an effect of nutritional status but also a manifestation of increased exposure to factors generating oxidative stress and inflammation of the nervous tissue [18,38,40], and a sign of the presence of efficient protective mechanisms of this tissue from damage.

As mentioned earlier, the Vgs concentration in foragers is formed primarily by environmental factors, affecting the bee colony strength. Therefore, positive and negative correlations between the Vgs level in all analyzed tissues of foragers and the colony strength were present (Table 4). Most such correlations were found for Vgs in the entire bodies of foragers (positive and negative correlations, *R* = 1). The Vgs function explains this relationship—supporting the functioning of the immune system and protecting against oxidative stress [15,17,18,19,39].

High Vgs levels indicate good health and more extended bee life [20]. A strong correlation between the Vgs level, primarily in the entire bodies of foragers, and the bee colony strength can indicate a compensatory mechanism of the Vgs synthesis to the colony strength. A reduction in Vgs concentration was observed in colonies with many insects the following month. For instance, in the U1 colony, the most numerous bee colony in the urban apiary, an average of 56,385 insects were found in July. In the following month, there was a 1.3-fold reduction in Vgs in the fat bodies and the entire body of the foragers from this colony, and a 1.4-fold reduction in their brains compared to the previous month. The opposite relationship was found in weaker colonies. For instance, an average of 14,894 bees were found in the U5 colony from the urban apiary in May. In the following month, an increase in Vgs concentration was shown in all analyzed tissues of foragers from this colony, compared to the previous month (10.4-fold increase in the fat body, 1.5-fold increase in the brain, and twofold increase in tissues from the entire body of bees). The validity of the demonstrated theory about compensatory mechanism results from the fact that the presence of high levels of Vgs has a positive effect on the life expectancy of the workers.

The nutritional role of Vgs and its positive effect on resistance to stressors such as oxidative stress, hunger, and infections can explain this effect. Hence, Vgs are believed to be the primary regulator of bee lifespan [12,16,18,39]. Therefore, according to our theory, the Vgs level increases in the following month, immediately after the month with weak colony strength. Such variability in Vgs level can constitute a compensatory mechanism in not numerous bee colonies. The role of the compensatory mechanism is to extend the bee life and thus ensure the appropriate number of foragers responsible for providing nectar, pollen, and water for the colony. Notably, an increase in Vgs in foragers positively affects the preference for collecting pollen instead of nectar [16]. Pollen is essential for developing the pharyngeal glands in nurses responsible for producing food for the larvae. Without adequate pollen in the bee colony, larval cannibalism occurs [45,46,47]. Moreover, the increase in Vgs concentration inhibits feeding activities and thus promotes, among others, behaviours related to brood care [12,16,48]. The influence of Vgs on these worker behaviors makes it possible to provide the food necessary to raise the next generation of bees. However, in numerous colonies, there is no need to introduce compensation mechanisms—extending the life of foragers, shifting preferences to collecting pollen, or transitioning to a subcaste of workers taking care of the brood. Hence, a reduction in Vgs level in foragers from such huge colonies is observed.

The proposed mechanism of bee colony plasticity has not yet been demonstrated in other studies. The described relationship between the colony strength and the Vgs level is more pronounced in foragers from urban apiaries. As mentioned earlier, it can result from increased exposure to factors generating oxidative stress and inflammation and may also result from a better nutritional status. The high level of Vgs can be a biomarker of bee colony depopulation. Using the Vgs level as a biomarker of the honeybee colony strength can predict the fate of colonies, especially those with low numbers. The compensation mechanism during periods of worker deficiency in the bee colony can explain the evolutionary success of social insects such as honeybees.

## 5. Conclusions

Honeybee colonies from the urban apiary were stronger than colonies from the rural apiary. The bee colony strength depends primarily on environmental and not genetic factors. The features of the urban ecosystem seem to play a crucial role in forming the colony strength and health. The relationship between the Vgs level in the entire body of foragers and the colony strength can be the compensation mechanism during periods of worker deficiency in the bee colony. The proposed compensatory mechanism can be one of the reasons for the evolutionary success of honeybees. It was more pronounced in foragers from urban apiaries. The high level of Vgs can be a candidate for bee colony depopulation biomarker. Using the vitellogenin level as a biomarker of the honeybee colony strength can predict the fate of colonies, especially those with low numbers.

## Figures and Tables

**Figure 1 insects-16-00025-f001:**
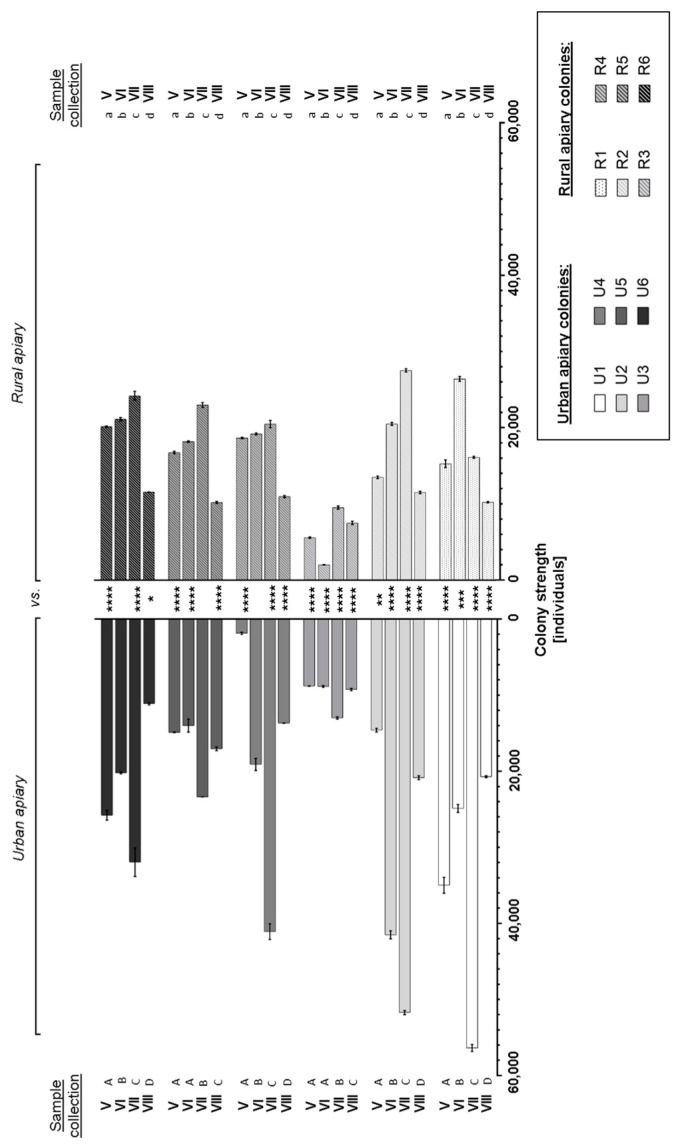
The honeybee colony strength from the urban (U1–U6) and rural (R1–R6) apiaries expressed by the number of individuals [mean ± SD] in the months—from May (V) to August (VIII). Tukey and Sidak multiple comparison tests, *p* ≤ 0.05. Asterisks (*)—statistically significant differences between paired colonies for a given month are marked with the same shade of grey (*—*p* ≤ 0.05, **—*p* ≤ 0.01, ***—*p* ≤ 0.001, ****—*p* ≤ 0.0001). Different letters—statistically significant differences within one colony over the analyzed season (capital letters—urban apiary, lowercase letters—rural apiary).

**Figure 2 insects-16-00025-f002:**
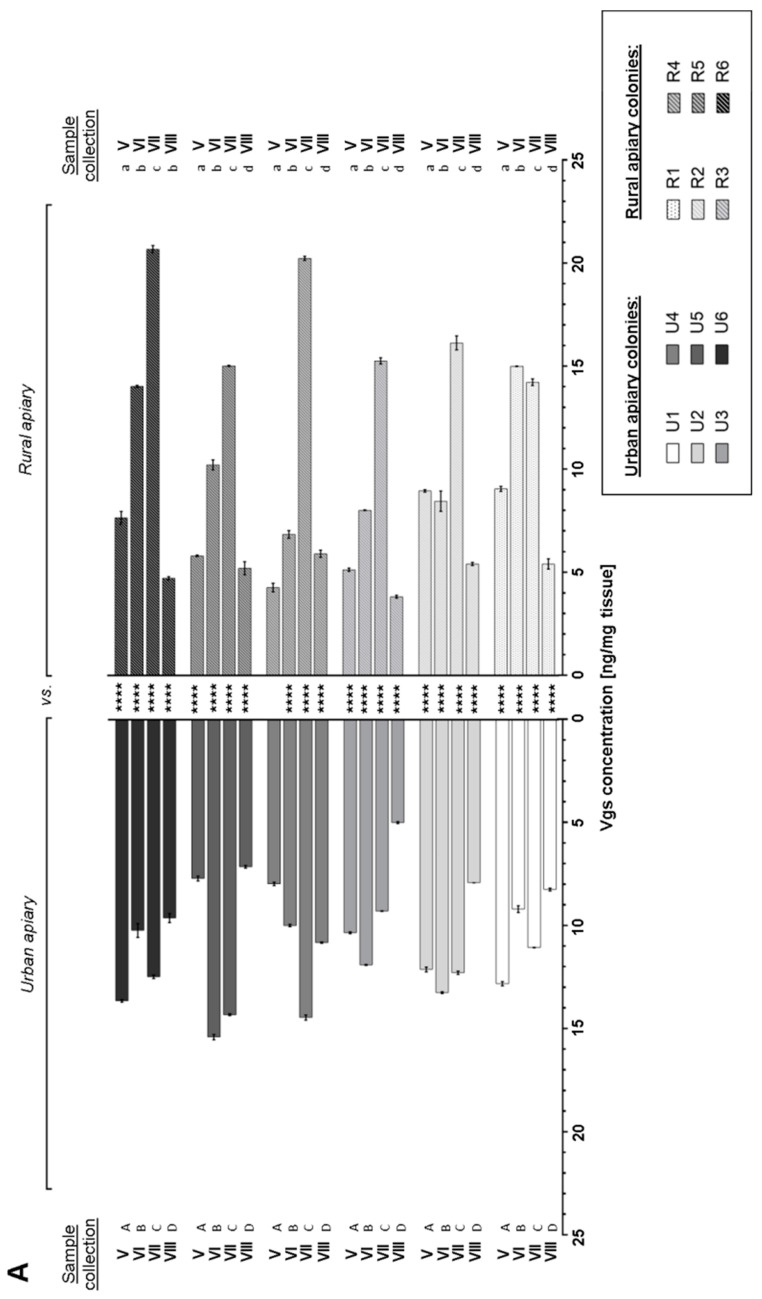
Vitellogenins (Vgs) concentration [mean ± SD] expressed in ng/mg of tissue in the entire body (**A**), brain (**B**), and fat body (**C**) of *Apis mellifera carnica* foragers from the urban (U1–U6) and rural (R1–R6) apiaries collected from May (V) to August (VIII). Tukey and Sidak multiple comparison tests, *p* ≤ 0.05. Asterisks (*)—statistically significant differences between paired colonies for a given month are marked with the same shade of grey (****—*p* ≤ 0.0001). Different letters—statistically significant differences within one colony over the analyzed season (capital letters—urban apiary, lowercase letters—rural apiary).

**Table 1 insects-16-00025-t001:** MANOVA analysis with the apiary localization (urban vs. rural), subsequent colonies (from 1 to 6) and the data of material collection as sources of variation in the strength of bee colonies. Statistically significant *p* values (*p* ≤ 0.05) are marked in red.

Factors	df	F	*p*
Localization	1	9634.2	0.001
Subsequent colonies	5	5866.5	0.001
Data of sample collection	3	9022.7	0.001
Localization × Subsequent colonies	5	1716.0	0.001
Localization × Data of sample collection	3	2122.9	0.001
Subsequent colonies × Data of sample collection	15	805.4	0.001
Localization × Subsequent colonies × Data of sample collection	15	795.6	0.001

**Table 2 insects-16-00025-t002:** MANOVA analysis with the apiary localization (urban vs. rural), subsequent colonies (from 1 to 6), the month of material collection, and the tissue/organ analyzed as sources of variation in the vitellogenins concentration in *Apis mellifera carnica* foragers. Statistically significant *p* values (*p* ≤ 0.05) are marked in red.

Factors	df	F	*p*
Localization	1	2.9	0.0008
Subsequent colonies	5	194.6	0.0000001
Data of material collection	3	4339.4	0.0000001
Tissue	2	15,479.1	0.0000001
Localization × Subsequent colonies	5	54.4	0.0000001
Localization × Data of material collection	3	1495.7	0.0000001
Subsequent colonies × Data of material collection	15	141.3	0.0000001
Localization × Tissue	2	114.4	0.0000001
Subsequent colonies × Tissue	10	42.6	0.0000001
Data of material collection × Tissue	6	2387.0	0.0000001
Localization × Subsequent colonies × Data of material collection	15	113.5	0.0000001
Localization × Subsequent colonies × Tissue	10	44.6	0.0000001
Localization × Data of material collection × Tissue	6	240.5	0.0000001
Subsequent colonies × Data of material collection × Tissue	30	53.2	0.0000001
Localization × Subsequent colonies × Data of material collection × Tissue	30	48.9	0.0000001

**Table 3 insects-16-00025-t003:** The correlation coefficient r between the concentration of vitellogenins in the entire body (EB), brain (B), and fat body (FB) of *Apis mellifera carnica* foragers a from colonies (from 1 to 6) that was a source of bees in the same month (V-VIII). Pearson correlation, *p* ≤ 0.05. Colors mark the *r*-value that is statistically significant and in the range, indicating the presence of correlation (0.2 ≥ *r* ≥ 1). Color scale for assessing the correlation *r* value: 0–0.2—no correlation, 0.2–0.4—weak, 0.4–07—medium, 0.7–0.9—strong, 0.9–1—very strong.

			Vgs Concentration
			1	2	3	4	5	
			EB	B	FB	EB	B	FB	EB	B	FB	EB	B	FB	EB	B	FB	EB	B	FB
**Vgs concentration**	**V**	**EB**		0.15	−0.25		−0.71	0.46		0.95	0.8		−0.2	−0.5		0.38	0.78		−0.1	0.17
**B**	−0.42		−0.36	−0.74		0.29	−0.71		0.56	0.15		0.95	−0.49		0.88	−0.48		0.96
**FB**	−0.30	−0.98		0.99	−0.83		−0.71	−0.99		0.15	0.15		−0.85	0.88		−0.88	0.83	
**VI**	**EB**		0.74	0.15		0.15	0.01		−0.25	0.15		0.01	0.01		0.15	0.01		0.01	−0.25
**B**	0.25		0.15	0.15		0.98	0.15		1.00	0.01		0.51	0.15		−0.9	0.01		−0.8
**FB**	0.15	−1.00		0.15	−0.73		0.15	−0.87		0.01	0.53		0.15	0.99		−0.25	−0.83	
**VII**	**EB**		0.97	0.50		0.15	0.01		0.35	0.15		0.01	0.15		−0.15	0.01		0.01	−0.35
**B**	0.15		0.70	0.01		−0.94	0.15		1.00	0.15		−0.4	0.15		−0.9	0.01		0.98
**FB**	0.15	0.20		−0.35	−0.96		0.15	0.80		0.15	0.59		−0.25	−0.82		0.01	−0.93	
**VIII**	**EB**		0.15	−0.25		0.01	0.01		−0.25	0.15		0.15	0.15		1.00	1.00		0.01	0.01
**B**	1.00		1.00	−1.00		0.21	−1.00		−0.99	0.89		−0.9	0.15		0.15	−1.00		0.95
**FB**	0.10	0.25		0.15	0.15		−1.00	0.15		0.15	−0.93		0.15	0.55		0.15	0.15	

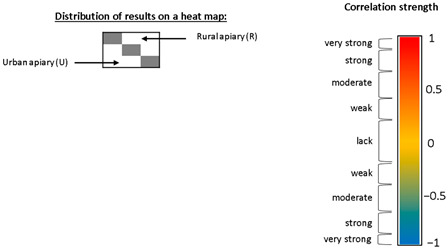

**Table 4 insects-16-00025-t004:** The correlation coefficient r between the concentration of vitellogenins in the entire body (EB), brain (B), and fat body (FB) of *Apis mellifera carnica* foragers and the colony strength (from 1 to 6) that was a source of bees in the same month (V-VIII) in urban (U; upper table) and rural (R, lower table) apiary. Pearson correlation, *p* ≤ 0.05. Colors mark the *r*-value that is statistically significant and in the range, indicating the presence of correlation (0.2 ≥ *r* ≥ 1). Color scale for assessing the correlation *r* value: 0–0.2—no correlation, 0.2–0.4—weak, 0.4–07—medium, 0.7–0.9—strong, 0.9–1—very strong.

		Vgs Concentration
		U1	U2	U3	U4	U5	U6
		EB	B	FB	EB	B	FB	EB	B	FB	EB	B	FB	EB	B	FB	EB	B	FB
**Colony strength**	**V**	0.10	0.10	0.12	0.00	0.02	0.00	1.00	0.10	0.10	0.01	0.10	1.00	0.15	0.00	−0.02	0.10	0.00	0.01
**VI**	−1.00	−0.01	0.10	1.00	0.10	0.02	−1.00	0.15	0.00	1.00	0.00	−0.02	1.00	−0.15	−0.20	1.00	0.15	0.02
**VII**	1.00	0.15	0.10	1.00	0.15	0.20	1.00	0.20	0.15	1.00	0.00	0.01	1.00	−0.01	−0.10	1.00	0.15	0.02
**VIII**	−1.00	−1.00	0.10	−1.00	1.00	0.15	−1.00	1.00	0.00	−1.00	0.02	−0.07	1.00	−0.02	−0.15	−1.00	1.00	0.02
		**Vgs Concentration**
		**R1**	**R2**	**R3**	**R4**	**R5**	**R6**
		**EB**	**B**	**FB**	**EB**	**B**	**FB**	**EB**	**B**	**FB**	**EB**	**B**	**FB**	**EB**	**B**	**FB**	**EB**	**B**	**FB**
**Colony strenght**	**V**	1	0.00	0.10	0.15	0.01	0.10	0.00	0.15	0.01	0.10	0.01	0.01	0.00	0.15	0.15	0.10	0.01	0.01
**VI**	0.1	0.01	−1.00	1.00	0.02	0.20	−1.00	0.20	0.20	1.00	0.10	0.00	−1.00	0.10	0.20	1.00	0.02	0.02
**VII**	0.1	0.02	0.25	−1.00	0.15	0.15	1.00	0.01	0.01	−1.00	0.20	0.02	1.00	0.00	0.10	1.00	0.15	0.15
**VIII**	1	0.02	0.20	1.00	0.20	0.15	−1.00	0.01	0.15	1.00	0.10	0.01	−1.00	0.00	0.12	1.00	0.20	0.20

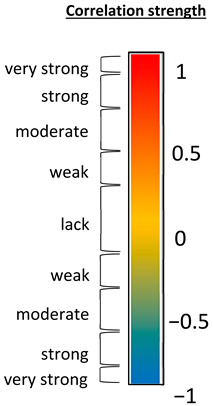

## Data Availability

The original data presented in the study are openly available in Zenodo, Vitellogenin level as a biomarker of the honeybee colony seasonal dynamics at DOI number 10.5281/zenodo.14168537.

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
