# Peer review of "Vitellogenins Level as a Biomarker of the Honeybee Colony Strength in Urban and Rural Conditions"

_insects, 2024, doi:10.3390/insects16010025_

Round 1

Reviewer 1 Report

Comments and Suggestions for Authors

The Authors' research is relevant to the growing popularity of urban apiaries.

However, in my opinion, the work needs significant modifications, especially with the way the data is analyzed and presented. This is especially true in the way the results are presented. After changing the way the data is analyzed and the results are presented, the discussion will have to change as well.

I’m wondering why the authors compared in pairs colonies from the rural environment with those from the city. Studies of this type usually compare group averages and medians. Box diagrams are very well suited for presenting such data. They simultaneously present many characteristics of the data. Current figures 1 and 2 can be placed in supplementary materials.

Also, correlations should be calculated within groups, not within individual colonies within a group.

It is also impossible to assess the size of the database, as the Authors did not provide the number of bees they examined from each colony. Why to evaluate the distribution of the data two tests were used, the Kolmogorov-Smirnov and Shapiro-Wilk tests? I guess that the database was not large, in which case Shapiro-Wilk tests are sufficient.

Throughout the work, the word “family” should be changed to “colony”.

Line 70 - The word “bred” should be replaced by the word “keept”.

Figure 1. - The R3 colony in June had a strength close to 0, how it later recovered? This is rather unlikely, probably the wrong data was entered.

Reviewer 2 Report

Comments and Suggestions for Authors

General Comments

This article has some useful data on the comparison of Urban Rural apiaries, but lacks support of floral data as well as vitellogenin in the insect body. First, Vitellogenin is a female specific reproductive protein. After its production in the fat bodies, this comes in the haemolymph that carries it to the developing oocytes. Therefore, study of  vitellogenin titer in the haemolymph should have been the most important aspect of this study which the authors have completely overlooked. Second, the title of the article is vitellogenin as a biomarker of colony seasonal dynamics. However, authors have made study for three months only and no seasonality is involved. Instead, they have shifted the whole contents to rural versus urban beekeeping. First of all, the authors should clarify as to what they mean by ‘Vittellogenin as a Biomarker’ keeping in view the fact that vitellogenin is a female specific reproductive protein in insects. Second, in my opinion the differences in the colony parameters are due to the differences in the floral conditions at two places. Authors have not provided the floral calendars of two places. Therefore, due to some pressing queries, I will have to recommend a major revision of this article.

Specific Comments

Title: Seeing the contents of the article, the title does not seem to be appropriate. Consider revising the title on the rural urban basis.

Abstract: Lines 21-23: This is not the subject of your study. Here, in the first 2-3 lines write the purpose of this study. Then in 3-4 lines briefly write the methods used. In next 4-5 lines write the salient results with some hard data if possible. In the last 2-3 lines write the conclusion with future projections.

Introduction

1.      Introduction is not satisfactory and needs thorough re-writing. In the first paragraph, authors should explain the purpose of this study. Then, in second and third paragraph review the pertinent literature. In the last paragraph, in the light of reviewed literature build your hypothesis, and then clearly define the objectives of this study.

2.      Line 40: Longevity of which caste, Queen or worker?

3.      Line 41-46: Whose immunity? Are these studies on worker bees or queen bee.

4.      Line 47: Not true; what the authors observe in Poland or Europe is not universal. In global south, floral-climatic conditions are entirely different from global north. So, rural urban divide cannot be generalized.

5.      Lines 47-52: These are redundant, not related to this study. Review the relevant literature.

Material and Methods

1.      Materials and Methods need to be elaborated to demonstrate which and how the colony parameters were recorded.

2.      Write the whole article in a Past Tense

3.      Write the age of the bees whose Vg was recorded.

4.      Line 70: bees were bred: Did you conduct breeding trials or you simply kept the bees in wooden hives. If the latter is true, then write that you simply managed the colonies as recommended in your region.

5.      Line 72: How many colonies were there in an apiary? Write here.

6.      Line 77: How many colonies were there in the Rural Apiary? Write here.

7.      Lines 104-121: You earlier wrote about Immunity, but did not measure the haemolymph vitellogenin. This seems something very strange. Perhaps Immunity should be measured in terms of haemolymph cells?

8.      Line 84: replace the word families with –colonies

9.      Line 99: How far the two apiaries were? Write in the previous paragraph. And what do you mean by ‘condition’? Did you measure the colony parameters?

10.   Line 105: Why did you exclude the haemolymph?

11.   Line 107: How could you measure the so small tissue with such an accuracy? Which measuring tool was used? Kindly write here.

12.   Line 116: How could you prepare the Standard Curve? I mean, which protein was used for this purpose, mention here.

13.   Line 119-120: Kindly write here how many observations were recorded for each sample.

14.   Here, in the last, write which colony parameters were compared. Say which were intra colony parameters, which were inter colony parameters, which were inter apiary parameters etc.

15.   Line 123: Not clear what did you do. Kindly clarify as to how many total observations were there?

Results

Results need to be more elaborated.

1.      Line 143: Not clear, what did you measure? Language of the Sentence is not very clear. Did you measure the total number of bees in the colony or did you measure the number of frames fully covered with bees. There are different methods of recording the colony strength.

2.      Fig 1 is not good representation of the data. I wish you present the urban rural figures side by side instead of up-down (This is for all the figures; kindly re-draw all for a better comparison) .

3.      Lines 147-148: There must be differences in the floral resources at the time of colony rearing. Floral conditions also need to be tabulated.

4.      Lines 166-182: Vitellogenins level: In this section, you showed the differences. Also mention which location was better?

5.      Lines 179-182: What these results signify?

6.      Supplementary data: 1. I could not understand these tables. But, what I could understand was that you have made comparisons between Rural and Urban Colonies. These comparisons are between colonies having same number as well as different numbers. If this is so, then presentation is not in order. The labeling ‘Raral Colonies’ in the vertical column is okay, but the labeling Urban Colonies should be at the top of horizontal bars. If my point is correct, then re-label these tables. If my point is not correct, then explain what you have done in these tables.

7.      2. These tables show differences but don’t show which apiary is better.

8.      3. Your study is only for three months. This is the floral period in Europe. But, without knowing the floral calendar of two places, it is difficult to say which site/place is better for the honey bee colonies.

Discussion

1.      Lines 283-297: These are the Results I wish to see these under the section Results.

2.      In the first 2  paras, authors have discussed the presence of Vg in different parts of the insect body and its higher or lower levels. In the later para they compared it between the apiaries and ultimately concluded the urban areas have better resources. Here I would differ. Again there are two issues; first about the time of recording of these observations. In those months urban areas may be having better floral resources. Second, this cannot be a generalization. As I pointed out elsewhere, global south is entirely different from global north. Global south is better for beekeeping during Oct-Feb; converse is true for global north. So, I will request the authors to carefully qualify the statements.

3.      Lines 301-307: The contents are not very clear. Do you talk about the strength of colonies or something else. From where this number has come. I do not find in Mat & Method that you recorded the number of bees anywhere.

4.      Lines 308-318: Again not very clear. Kindly clarify what you want to write.

Conclusion

1.      Line 327: replace the word ‘abundant’ with ‘strong’

2.      Line 328-329: Again qualify the statement. It is for global north.

3.      Lines 331-332: Argument seems exaggerated.

4.      Conclusion should be: ‘In the same months and under similar weather factors the urban colonies became stronger than the rural colonies. The bees from the urban colonies had higher level of vitallogenin than the bees from the rural colonies. Therefore, vitellogenin protein seemed to play a major role in augmentation of colony strength’.

References

Ok

Comments on the Quality of English Language

Some sentences are not understandable. Authors should read the article carefully to improve its readability.

Round 2

Reviewer 1 Report

Comments and Suggestions for Authors

Line 31 and 383 – “The high level 31 of Vgs can be a biomarker of bee colony depopulation”.

The authors did not investigate this aspect. This statement is not based on research.

I still do not know how many bees from each colony were used in the research.

How many bees from each colony in each term were used for evaluation vitellogenin concentration in: 1. body, 2. brain, 3. fat body?

Whether 3 vitellogenin concentration analyses were performed from the same bee – body, brain and fat body?

Line 167 – “All analytic determinations were carried out in triplicate for each of the six samples from each tissue (total observation for each tissue = 18)”

This information should be in the methodology.

Does this mean that each type of analysis from each bee was performed 3 times - 3 samples from one bee, or 3 analyses from one sample – one bee?

 I maintain all previous comments regarding the analysis and presentation of data:

“However, in my opinion, the work needs significant modifications, especially with the way the data is analyzed and presented. This is especially true in the way the results are presented. After changing the way the data is analyzed and the results are presented, the discussion will have to change as well.

I’m wondering why the authors compared in pairs colonies from the rural environment with those from the city. Studies of this type usually compare group averages and medians. Box diagrams are very well suited for presenting such data. They simultaneously present many characteristics of the data. Current figures 1 and 2 can be placed in supplementary materials.

Also, correlations should be calculated within groups, not within individual colonies within a group.”

The Authors' research does not concern toxicology but the influence of environmental factors on bee colonies. Such research is conducted in the aspect of the population, not a single colony.

This is the first time I have encountered the opinion presented by the Authors, and I have been engaged in similar research for 24 years.

What principle was followed when selecting colonies into pairs?

Medrzycki, P. et al. Standard methods for toxicology research in Apis mellifera. J. Apic. Res. 52, 1–60 (2013) - please indicate the exact recommendations that I should pay attention to.

Reviewer 2 Report

Comments and Suggestions for Authors

Dear authors, It is well understood that a reviewers may not be fully conversant with the research conducted by the authors, but their work is to carefully examine the presented research and find faults (if any) to make the article suitable for publication and acceptable to the scientific community (as much as possible). And you will agree that the revised version is a better presentation.  However, you may re-examine the Tables in the supplementary material. There is a set method of presentation of a Table. There are Columns and Rows. In the column you write the number of colony in (say) Urban Apiary, and in Rows you write the same numbers of (say) Rural apiary. Then you can write the values against each pair. This way you can revise the Table. 

I am happy that you have revised your article satisfactorily. 

Author Response

Dear Reviewer,
The Authors would like to thank the Reviewer for the comments and suggestions that helped correct the manuscript. The authors are pleased that they met the Reviewer's requirements and that the Reviewer accepted all the corrections introduced. 

Due to some ambiguities in the interpretation of tables included in the supplementary materials (Table S1, S2), we have introduced some editorial changes. The authors hope that they will now be clear to understand.